# Amelioration of Pet Overpopulation and Abandonment Using Control of Breeding and Sale, and Compulsory Owner Liability Insurance

**DOI:** 10.3390/ani11020524

**Published:** 2021-02-18

**Authors:** Eva Bernete Perdomo, Jorge E. Araña Padilla, Siegfried Dewitte

**Affiliations:** 1University of Las Palmas de Gran Canaria, 35017 Las Palmas de Gran Canaria, Spain; jorge.arana@ulpgc.es; 2Katholieke Universiteit Leuven, 3000 Leuven, Belgium; siegfried.dewitte@kuleuven.be

**Keywords:** pet abandonment, pet overpopulation, exotic pets, pet market regulation, pet welfare

## Abstract

**Simple Summary:**

Overpopulation and abandonment of pets are long-standing and burgeoning concerns that involve uncontrolled breeding and selling, illegal trafficking, overpopulation, and pet-safety and well-being issues. Historical and current prevention measures for avoiding these problems, such as sanctions, taxes, or responsibility education, have failed to provide significant moderation or resolution. Globally, millions of pets are commercially and privately bred and abandoned annually, damaging biodiversity and ecosystems, and presenting road safety and public health risks, in addition to becoming victims of hardship, abuse, and illegal trafficking, especially in the case of exotic species. This article proposes a novel comprehensive management system for amelioration of overpopulation and abandonment of pets by using greater control of supply and demand of the pet market, highlighting the role of the compulsory owner liability insurance to prevent pet abandonment and all its associated costs. This system aims to act preventatively, through flexible protocols within the proposed management system to be applied to any pet and any country.

**Abstract:**

Overpopulation and abandonment of pets are long-standing and burgeoning concerns that involve uncontrolled breeding and selling, illegal trafficking, overpopulation, and pet safety and well-being issues. Abandonment of pets creates numerous negative externalities and multimillion-dollar costs, in addition to severe consequences and problems concerning animal welfare (e.g., starvation, untreated disease, climatic extremes, uncertainty of rescue and adoption), ecological (e.g., invasive species and introduction of novel pathogens), public health and safety (e.g., risks to people from bites, zoonoses, or road hazards), and economic (e.g., financial burdens for governmental and nongovernmental organizations). These interwoven problems persist for several reasons, including the following: (1) lack of an efficient system for the prevention of abandonment and overpopulation, (2) lack of regulatory liability for pet owners, (3) lack of legal alternative to abandonment. This article proposes a novel comprehensive management system for amelioration of overpopulation and abandonment of pets aimed to tackle the current supply and demand dysfunction of the pet market and provide a legal alternative to abandonment.

## 1. Introduction

Overpopulation and abandonment of companion animals is a historical, endemic, and systemic worldwide problem, which has worsened in recent years with the economic recession [1,2] and the growth of the international legal and illegal pet trade [3,4], among other factors. The consequences of overpopulation and abandonment include at least four factors:Animal welfare—Animals may face novel hardships, including starvation, untreated disease, climatic extremes, uncertainty of rescue, and if rescued, uncertainty of adoption [5,6,7,8,9]. Consequently, many or most such animals are euthanized due to lack of space and resources [10,11]. Relatedly, some governments have established rules on zero sacrifice [12,13,14], which can become inoperable when pets accumulate unmanageably. Ecological impact—Animals, especially exotic species, can become invasive organisms, which may cause damage to local wildlife through predation and disease transmission, and result in disruption to ecosystems [15,16,17,18]. Public health and safety—Animals may present risks to people from bites, zoonoses, or become road hazards) [19,20,21,22]. Economic—Animals can become financial burdens for governmental and nongovernmental organizations that assume their care [23,24,25,26,27,28]. 


We consider that these persistent consequences are interwoven with a lack of an efficient system for the prevention of abandonment and overpopulation, lack of a regulatory liability for pet owners and lack of legal alternative to abandonment. 

The principles of supply and demand establish that market economies must be balanced and stable [29]. However, pet markets lack these normal dynamics of economic balance and stability. 

In relation to demand, the current market for pets is undergoing a significant process of expansion, given the growing incorporation of animals into homes around the world [3,30]. In Europe, 50% of the population lives with at least one pet in their home [31], and in the United States, this proportion is 67% of households [32]. This demand lacks adequate control mechanisms to prevent animals from being abandoned and their owners deriving their obligations to third parties in the shape of negative externality, as will be explained in Section 2.2. 

In terms of supply, the pet market is becoming increasingly lucrative, which historical and current regulatory mechanisms have failed to moderate or even adequate monitor [33,34,35,36]. Accordingly, most commercial pet suppliers appear to be subject to few controls [3], and a diversity of species (at least 13,000 [37]) pervade the market system. Large numbers of individual exotic animals are also involved (estimated to be at least 350 million per year [38]), and the world population of dogs is estimated to currently stand at more than one billion [39]. 

A result of these issues is that the number of pets that enter the market far exceeds the demand, which translates into overpopulation. In accordance with general economic principles, commodification and overabundance of animals depresses both price and value [3,40,41]. When an owner–pet bond suffers reduced value, the probability of abandonment and euthanasia are high [42]. Also, overpopulation can result in pets being freely given away or even attract “inverse prices”, where owners pay others to take on their animals. 

Essentially, at least some of the origins and causes of overpopulation and abandonment of pets on a global basis derive from asymmetry and imbalance of supply and demand. In the following sections, we summarize the problems inherent to the overpopulation and abandonment of pets, including the underlying causes, economic costs, and a novel management system concept to ameliorate several established problems. 

## 2. Discussion

### 2.1. Historical and Current Management of Pet Overpopulation and Abandonment

Historical and current management of pet overpopulation and abandonment presents the following characteristics:1.Implementation of penalties or criminal sanctions [12,13,14,43,44,45]—In practice, this measure is minimally effective, as evidenced by increasing incidences of abandonment [46,47,48,49,50,51]. Abandonment is usually a covert and anonymous act, making location and punishment of offenders difficult. 2.Legislation usually prescribes for identification of animals using microchips [52,53,54]—While microchips allow for identification of animals and owners, and thus avoidance of them being discarded on, for example, public roads, this approach does not prevent delivery of animals to shelters or kennels, which effectively constitutes abandonment. Owners may also abandon animals and claim they were lost, accidentally released, or that there were problems with the microchip data [55,56,57,58,59,60]. 3.Establishment of fees or taxes on the keeping of pets, which are used only in some countries with little success (based on revenue collection), rather than ending overpopulation or neglect [61,62,63]. 4.Lack of legal alternative to abandonment—Pet owners may feel pressure to abandon when they face or experience unforeseen or unpredictable situations such as loss of employment, changes of residence, family health, pet behavior problems, incorrect expectations about pet owners’ responsibilities, pet biological needs, and feel they can no longer provide adequate care [64,65,66,67,68,69,70,71,72]. Although national laws may prohibit abandoning a pet, personal, family, economic or other circumstances may point to abandonment as the most feasible solution. This conflict of interests appears not to be legislated for and has therefore remained unresolved. Because of these applied legislative deficiencies, owner and state responsibilities are de facto shared, yet dictated by the owner’s personal situation and his ethical and moral principles. In our view, ameliorating pet overpopulation and abandonment requires fundamental shifts in promoting owner understanding of their responsibilities, as well as changes in the current management of breeding and sale. 5.Lack of a comprehensive management system to address overpopulation and abandonment. The aforementioned characteristics lack of connection and coordination among them, and this is essential to tackle the problem properly.


### 2.2. Economic Factors and Costs of Overpopulation and Abandonment of Pets: Negative Externalities

In economics, negative externalities imply the costs associated with the activities of an individual or collective that cause significant and ongoing expenses to others [73]. Overpopulation and abandonment of pets worldwide generate multimillion-dollar costs that are commonly met by governmental and nongovernmental sectors rather than by the private owners responsible for their release. 

Global numbers for abandoned animals are unavailable, and present data are estimated (probably underestimated [5,6,7,8,9]) and comprise only animals entering governmental and nongovernmental shelters. Wider assessment could include unrecorded morbidity or accident-related mortalities prior to rescue. Available figures indicate that in the United States 4 million dogs are abandoned each year [74], in Spain more than 100,000, [75], in the United Kingdom, 130,000 [1], and in Australia, 250,000 [9]. A diversity of exotic pets that are also abandoned must be added to this situation [64,76]. There are no official data or studies on abandonment of pet amphibians, reptiles, birds, or unusual mammals, but existing data on pet releases or escapes probably also underestimate the numbers of released or escaped exotic pets [77]. 

To summarize and improve the understanding of the economic impact incurred by the abandonment of pets, we can disaggregate the cost structure into two primary categories, according to their consequences: 

Category A—costs associated with pets that have been abandoned and are rescued promptly, which produces direct costs related to the necessary attention, care, and maintenance provided by shelters and kennels. For example, a small dog with an average life expectancy of 16 years [78] is assumed abandoned and rescued within one year. According to studies carried out on abandonment and costs, the approximate average daily cost of keeping this dog is €6 for Spain, [75,79], the United Kingdom [80], Australia [9], and the USA [74]. During the lifetime of the dog, these costs would amount to €34,560 for governmental or nongovernmental sources, assuming it is not adopted or sacrificed, which occurs in 60–70% of cases [10,11]. Maintenance of abandoned exotic pets requires more expensive and complex care and represents a considerable burden on pet shelters [23,24,25,26,27,28]. 

Category B—costs caused by pets that have been abandoned but could not be rescued. These pets produce incalculable indirect costs, including those related to the following:1.Animal-associated traffic accidents, in which both people and pets are victims. These costs can amount to up to £14.6 million annually in the UK [81,82] and are one of the frequent causes of mortality in abandoned pets in urban areas [22,83]. 2.Negative impacts on biodiversity and ecosystems due to predation and disease transmission. Abandoned cats and dogs can present a threat to livestock and place native fauna at risk [84,85,86]. Abandoned or escaped exotic pets can impact local fauna due to predation and disease transmission and also cause irreversible damage to agriculture [87]. Once established, eradication of exotic species is difficult and expensive. Currently, invasive species affect all countries [48,88]. In Europe, the economic cost from damage caused by invasive species (animals and plants) has been estimated at approximately €12.5 billion per year, although this figure may represent only 10% of the potential cost [89,90]. In the United States, economic losses caused by invasive alien species are estimated at a minimum of $120 billion per year [91]. 3.Pets are associated with more than 60 zoonotic diseases [92,93,94,95,96], most of which are linked to exotic pets. Dogs are carriers of several potentially debilitating and dangerous diseases, including rabies, which is endemic is parts of Europe [95,96], the United States [97], and Asia [98]). In the U.K. alone, the annual cost of treating zoonotic diseases transmitted from dogs to humans is around £10 million per year, and the cost of dog attacks on humans is £4 million per year [82]. Exotic pets can transmit infections through stings, bites, or simple physical contact, potentially resulting in serious injury, disease, or even death [99,100]. Estimated costs for treating injury or infection caused by abandoned exotic pets are difficult to quantify but range from €250 per medical visit to €2500 per day of hospitalization [101]. The trade and keeping of exotic pets are recognized as a significant factor in the emergence and spread of zoonotic diseases [102]. Exotic pets are estimated to cause 60–75% of emerging diseases [48,103,104,105,106,107,108], including avian influenza and psittacosis from birds; salmonellosis from amphibians, reptiles, and birds; and hepatitis A, tuberculosis, monkey pox, and herpesvirus simiae-B from primates. SARS-Cov-2 (COVID-19) is thought to have been transmitted from wild animals to humans in wildlife markets in China [109] and the international trade in small carnivores [110]. In some cases, these disease introduction risks are increased by the illegal trade in pets that are smuggled into countries by circumventing border controls [111], although for most invertebrates, fishes, amphibians, and reptiles, no border quarantine is applied. Cumulatively, the management costs of these diseases incur trillions of dollars, to which the many human losses of life must be added. 


## 3. Recommendations

The system proposed below has been specifically designed to ameliorate pet overpopulation and abandonment and all associated costs. It is composed of the following elements:

### 3.1. Ameliorating Overpopulation and Abandonment Using Control of Breeding and Sale 

Currently, demand generates abandonment because, basically, pets are sold without external controls or verifications before the acquisition of the pet is formalized, which can lead to impulsive or ill-considered pet acquisitions. Information efforts targeted at pet ownership responsibilities such as the EMODE Pet Score System [112] may offer a useful part in self-education. All issues related to training, information, and advice to pet owners will be regulated by a training protocol specifically designed for this purpose. 

Because supply dysfunction produces overpopulation, actions aimed at resolving supply-based problems effectively infer controlling the breeding and sale of pets, and their promotion. Pet breeding and sale should only occur at authorized centers and monitoring and controlling this sector is vital to restore the balance of supply. 

To reduce health risks, breeding operations should have a veterinary team that guarantees that pets available for sale are free from infections caused by bacteria, viruses, or fungi that can be transmitted to other animals or people. 

To acquire a pet, people would require accreditation of suitability (e.g., regarding valid training, criminal record check, economic stability, availability of space, and other resources), and thus would not be able to buy it directly from stores, breeders, or online suppliers. Such prior conditions for ownership would also assist to set the rate of the production of pets, which would help prevent oversupply and thus overpopulation. 

### 3.2. Improving Owners Responsability Using Compulsory Owner Health and Survival Liability Insurance

Because demand dysfunction produces abandonment, balancing demand requires changing mindset responsibilities among actual and prospective pet owners. Our primary proposed measure for promotion of owner responsibilities (e.g., where they face unforeseen situations), and to prevent abandonment, as well as to better provide for pet welfare, is a novel compulsory health and survival insurance (CHSI) scheme. Such insurance would necessarily be arranged as a precautionary measure prior to pet acquisition, although post-acquisition CHSI should also be encouraged. 

CHSI could operate globally to guarantee essential services that each country considers appropriate for pets and their management when their owners, for whatever reason, cannot continue to care for them. As far as we are aware, CHSI does not exist anywhere in the world, and is the cornerstone for the prevention system of pet abandonment here proposed. 

CHSI would, for example, offer specific benefits to owners and pets, by guaranteeing costs in relation to:1.Owners who, for any reason or unexpected circumstance, cannot continue to care for their pet can contact an insurer-approved pet shelter (under veterinary supervision and with a general no-kill policy) for collection. 2.Pets can spend the rest of their lives in an approved shelter or be adopted under supervision. 3.Inclusive civil liability insurance against personal and material damage arising from the ownership of the pet. 4.Costs for lifelong pet support are assured, thus avoiding costs to the public purse. 


Relatedly, certain supportive efforts could assist CHSI measures, including the following: a shared ownership regime—sharing the pet with other owners to, for example, help in cases of travel/vacation cover or illness and cost division; and a formalized adoption—handing over an animal to a new owner who will have animal care costs met under the previous owner’s insurance policy. 

A feasibility study has not been conducted for the implementation of the proposed CHSI scheme because the costs would be strongly dependent on the situation in each country and the degree of commitment assumed by each government to end overpopulation and abandonment. However, nowadays, few would probably question the effectiveness of insurance to address negative externalities. If one considers the current costs to society of overpopulation and abandonment, then the CHSI scheme may be regarded as modest. For example, the estimated costs of maintaining a medium-sized dog in a pet shelter are around EUR 6 per day. If a pet spends its entire life in a shelter (assuming an average life expectancy of 16 years), the total cost will be EUR 34,560. In other words, in the event that the pet ceases to be cared for by its owner during the first year of life, the compensation provided by the insurance to the pet shelter would be around EUR 34,000. This indicates that the price of the risk premium could be around EUR 400–500 per year, for an average medium-sized dog. Therefore, we highlight that the main attribute of this insurance is that it helps to prevent causes and consequences of abandonment (and all its associated costs), regardless of the reason and the circumstances under which it may occur. 

### 3.3. Fostering Pet Welfare through Institutional, Legal, and Executive Guardianship

In order to try to end with the historical precariousness of companion animals, a model of institutional, legal, and executive guardianship can be established, similar, for example, to that which already occurs for the protection of minors or women that are victims of domestic or gender violence [113,114,115,116]. 

## 4. Conclusions 

Historical and current overpopulation and abandonment prevention systems have been unable to solve these problems. Causes of abandonment appear to have unforeseen or unpredictable situations as common denominators, and it can be inferred that most owners do not acquire a pet with the intention of abandoning it, rather this arises from a lack of training and pertinent advice on responsible pet keeping and a lack of advance awareness of consequences of unforeseen situations throughout pet ownership. 

To address the first issue, mandatory specific and compulsory training courses on responsible pet ownership could provide essential knowledge to prospective and existing pet owners regarding legal and moral responsibilities, as well as animal care [65,66,67,68,69,70,117], which may also reduce impulsive acquisitions. To address the second issue, compulsory owner liability insurance provides guarantees against abandonment and prevents cost burdens to governmental and nongovernmental organizations, as well as other negative externalities. 

Efforts to correct the dysfunction of the pet market supply and therefore avoid overpopulation should focus on carrying out controls prior to the acquisition of pets and the implementation of pet-ownership suitability protocols. 

The implementation of the proposed comprehensive management system for overpopulation and abandonment will create a new pet market completely different from the current one. The main challenge of the transition and convergence process between both markets is how to resolve the situation of the millions of animals that are currently abandoned. This process will be described in future publications. 

## Data Availability

Data sharing is not applicable to this article.

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
