# Peer review of "Amelioration of Pet Overpopulation and Abandonment Using Control of Breeding and Sale, and Compulsory Owner Liability Insurance"

_animals, 2021, doi:10.3390/ani11020524_

Round 1

Reviewer 1 Report

Review for Eradication of overpopulation and abandonment 2 through regulatory policies of the pet market

COMMENTS LINE BY LINE

  1. This paper starts with a simple summary that makes a big claim – that overpopulation and abandonment occurs because the systems rely on the wrong solutions. Let us see if this paper convinces me of this. (Having read the paper, the claim in not made out but the proposed philosophical claims are interesting)
  2. Line 22 – this point 5 does not stand alone, the other 4 seem to?
  3. Line 30 – the extra ‘with’ first word needs removing
  4. Line 53 - I cannot see where paper 5 makes the claim that 83% of the world’s dogs are abandoned?
  5. Line 76 – what is the outdated line of research? I guess the three points on lines 79 – 90?
  6. Line 85 – ah so you suggest changing philosophical doctrines? I agree these are a big part of the huge issue of overpopulation and abandonment (which I am going to call O&A from now on.
  7. Lines 92 – 96. I think what you are saying is that currently there are no laws to prevent owners abandoning their animals? Or are you saying there are laws, but pet owners will abandon because it is easy to do so as there is little fear of being caught? I like this paragraph but its not worded clearly. Also, you talk about an analysis of legislation – oh my there are what 195 countries, and within those countries there are subsets of law as well. So I am not sure you can make a statement about an analysis when really such a thing cant happen globally as each jurisdiction has such a different approach in attempting to address O&A, if they have a approach at all.
  8. Lin 113 - Does the law prevent abandonment? What law, which country and some countries do allow it, there is no problem is Australia taking an unwanted do to the pound – this is abandoning the pet, its not illegal.
  9. Lines 120 to 125 – would not this be lovely to achieve
  10. Line 128 – I am not sure if AD HOC is the best word perhaps more a flexible system?
  11. Line 137 – but it is not the same for all animals. Let us look just dogs. Some have little to no value and something with no value – who pays to protect and care? A five-year-old pit bulldog, bred by an accidental or occasional breeder that keeps one and abandons the rest? How can we control such a breeder? So, like with any product – some dogs have high value, some have none, we cannot confuse this – but all dogs have the same worth as dogs. This is the crux.
  12. Lines 142 – 151 – I would like some literature here; can we really say that this pure economic theory applies to the ‘economies’ of all animals? Doesn’t it rest on some principles that do not hold with animal breeding – that some people do it accidentally, some do it just for themselves? What other industries that produce a product – is built on such different underlying principles?
  13. Line 158 – this is a huge claim - are you saying that no one can own an undesexed rabbit and give or sell the babies? What about chickens for eggs? Hobby dog breeders? Bird? All of these animals are easy to breed – no one will ever stop people living with and breeding animals so your statement about

Breeding can only be carried 158 out in authorized centers, and outside these centers, the reproduction of pets must be absolutely 159 prohibited.

 And

Pets that 161 were born after the entry into force of the new regulations cannot be found without coming from 162 these authorized centers. Any attempt to reproduce outside these centers must be prevented through 163 severe administrative and criminal sanctions that have an effective dissuasive function

               Might be considered an impossibility to achieve. Your paper is an academic one I appreciate          that, but I seek some evidence here that even in any jurisdiction in the world such a thing has               worked?

  1. Line 164 – monitoring is going to be a huge cost – again you need evidence that this is economically feasible?
  2. Lines 173 – 184, you set out some good ideas but again you need some more detail, literature that supports its viability.
  3. Line 184. I think you should start a new paragraph as you move to considering “In order to acquire a pet…’ This part of your model is to my mind perhaps more achievable.  Again, there is literature out there that does confirm that in some countries – there are regulations that mandate a pet ownership licence before a pet is bought. You should put these types of examples in here.
  4. Lines 201 – 203 – are you suggesting this for the breeding of all animals? Even say zebra finches and mouse? These animals are sold for a few dollars they are so easy to breed and do not like a long life. How can you really suggest you will stop mice breeders, kids in bedrooms? We cannot control everything can we? What countries in the world are developed enough to put the money into creating this system that you propose? I would like to see some literature, where any country has attempted any part of what you are suggesting.
  5. Lines 225 – 241. Some good points raised here. Kids own pets, so many different types of pet, how will any country create and sustain such training?
  6. Line 244 – 245 – your new model of responsibility – that is a good term – but such a model has to start with education and cultural shifts? Have you read any work on responsive regulation and the role in any regulatory system of education and training? We cannot, can we go straight to regulation and enforcement when so many people are doing the right thing? Wouldn’t it be better to spend money on identifying and educating those that do the wrong thing? Even work by Elizabeth Tyson on Licensing Laws and Animal Welfare might be a useful thing to consider here?
  7. Lines 281 to 299 – your concept of H& S insurance. This is an interesting concept. But again, and I am sorry, I keep coming back to the economic realities of this. A wealthy businessman living in an apartment in New York, purchasing a $600O French Bulldog might see the sense of such insurance and it might work here but what about a mouse owned by a small child in a village. A child that needs to learn about caring for animals as his family are farmers. Can we really see that mouse being insured? If so, can you put in some articles where we do now insurance has helped. Pet insurance does exist so let us start with looking at that. Put some detail about it in here and if there are articles that confirm it is a good thing and offers a viable business to anyone, put that detail in and expand.
  8. Lines 321 – 340 - your comparison to the car industry insurance is a good one. The costs of pet care in a shelter can is not disputed. Would your argument, for this whole paper be better served by making a case, as you do here with dogs? At least some dogs do have enough value to warrant a consideration of making owners pay insurance, as I said about, many already do. Its too much to attempt to argue we can insure all animals. Perhaps propose it if for dogs in one or two countries and then make statements about the possibility of it moving into other animal breeding industries?????
  9. Lines 356 to 365 – yes good points it might but NOT FOR ALL ANIMALS? Until all people value the lives of all animals equally, until we are educated about the value of animal life and until we stop being human centric what you have is a lovely philosophical position  - not the framework for a viable insurance business or not the framework for a viable legal framework for the 195 countries in the world? If you have literature that disproves my view here, please pop it in.
  10. Lines 365 to 380 – I sound like the people your paper mentions indirectly (as always happens). I am not, I believe that the notion of insurance is a good one. It is the scope of your innovative idea – it needs to be smaller, its too big for 2020. Such a solution could be piloted with one animal and one country??
  11. Line 402 – I know in many countries and let us use Australia – many states, such as the state of Victoria have attempted to tackle O&A for many years. I know you quote work from Australia so you will have researched it. Even in states that have attempted to tackle it, animals are still being abandoned. I think you need examples of what good has been obtained where legislation in the form that most closely aligns to what you are proposing has achieved? Can you find any?
  12. Line 426 – yes, the creation of bodies that are there to protect animals and then to facilitate legislation at national level is a commendable thing – you need examples. Some countries have this, but can all countries really establish special protection and a push towards the emancipation of animals? Where is the political good will do this? Provide some examples where it does exist and go from there?
  13. Line 443 – who will pay for the de sexing? There are studies that look at this are not there.
  14. Line 455 – you propose price setting but by whom? Previously in the paper you have advocated the use of market forces to help regulate animals. Now you propose some agency become involved. How will it work in reality?
  15. Line 476 – you acknowledge no feasibility study has been carried out – but couldn’t you at least propose a country where something as close to what you propose is occurring or a country that you believe might be able to have some of these measures considered?

OVERALL COMMENTS

The paper was interesting to read.

From reading my specific line by line comments it is apparent that I do not believe that the underlying model that is being proposed by this paper is feasible.

I do however like the notion of Health and Survival insurance. If this paper could be rewritten with it not making such huge attempts but with the insurance as its main thrust, a concept that regulatory agents that regulate companion animal breeding might need to consider, I would support it. If this could be done and the other ideas reworded as things to aim for in an ideal country this paper would hold the reader’s interest. I think it needs to be supported with examples given of what of the ideas that have been proposed do at least exist in some countries. Give those examples, show how they can be extended. An example is the licensing of pet owners, then I think this paper might be interesting.

Author Response

The authors thank the reviewer for the time spent on the reviews and comments made. A complete restructuring of the article has been carried out so that most of the sections have been deleted or modified. Below we respond point by point to your comments.

Point 1: this point 5 does not stand alone, the other 4 seem to?

Response 1: This section has been deleted

Point 2: the extra 'with' first word needs removing

Response 2: This section has been deleted

Point 3: I cannot see where paper 5 makes the claim that 83% of the world's dogs are abandoned?

Response 3: Comment accepted. We have changed lines 114-122

Point 4:What is the outdated line of research? I guess the three points on lines 79 - 90

Response 4: This section has been deleted

Point 5:ah so you suggest changing philosophical doctrines? I agree these are a big part of the huge issue of overpopulation and abandonment (which i am going to call o & a from now on

Response 5:  This section has been deleted

Point 6: I think what you are saying is that currently there are no laws to prevent owners abandoning their animals? Or are you saying there are laws, but pet owners will abandon because it is easy to do so as there is little fear of being caught? I like this paragraph but its not worded clearly. Also, you talk about an analysis of legislation - oh my there are what 195 countries, and within those countries there are subsets of law as well. So i am not sure you can make a statement about an analysis when really such a thing can't happen globally as each jurisdiction has such a different approach in attempting to address o & a, if they have an approach at all

Response 6: We have changed  lines 93-101 to clarify this section.

Point 7: Does the law prevent abandonment? What law, which country and some countries do allow it, there is no problem is Australia taking an unwanted dog to the pound - this is abandoning the pet, its not illegal.

Response 7: This section has been deleted

Point 8: Would not this be lovely to achieve?

Response 8: This section has been deleted

Point 9: I am not sure if ad hoc is the best word perhaps more a flexible system?

Response 9: This section has been deleted

Point 10: But it is not the same for all animals. Let us look just dogs. Some have little to no value and something with no value - who pays to protect and care? A five-year-old pit bulldog, bred by an accidental or occasional breeder that keeps one and abandons the rest? How can we control such a breeder? So, like with any product - some dogs have high value, some have none, we cannot confuse this - but all dogs have the same worth as dogs. This is the crux

Response 10: This section has been deleted

Point 11: I would like some literature here; can we really say that this pure economic theory applies to the 'economies' of all animals? Doesn't it rest on some principles that do not hold with animal breeding - that some people do it accidentally, some do it just for themselves? What other industries that produce a product - is built on such different underlying principles?

Response 11: This section has been deleted

Point 12 This is a huge claim - are you saying that no one can own an undesexed rabbit and give or sell the babies? What about chickens for eggs? Hobby dog ​​breeders? Bird? All of these animals are easy to breed - no one will ever stop people living with and breeding animals so your statement about

Response 12:  We have changed  lines  173-218 to clarify this section.

Point 13: Monitoring is going to be a huge cost - again you need evidence that this is economically feasible?

Response 13:  We have changed  lines  173-218 to clarify this section.

Point 14: you set out some good ideas but again you need some more detail, literature that supports its viability. ?

Response 14: This section has been deleted

Point 15: I think you should start a new paragraph as you move to considering “in order to acquire a pet… 'this part of your model is to my mind perhaps more achievable. Again, there is literature out there that does confirm that in some countries - there are regulations that mandate a pet ownership license before a pet is bought. You should put these types of examples in here

Response 15: This section has been deleted

Point 16: are you suggesting this for the breeding of all animals? Even say zebra finches and mouse? These animals are sold for a few dollars they are so easy to breed and do not like a long life. How can you really suggest you will stop mice breeders, kids in bedrooms? We cannot control everything can we? What countries in the world are developed enough to put the money into creating this system that you propose? I would like to see some literature, where any country has attempted any part of what you are suggesting

Response 16: We have changed  lines  173-218 to clarify this section.

Point 17: some good points raised here. Kids own pets, so many different types of pet, how will any country create and sustain such training?

Response 17: The compulsory training on responsible pet ownership is aimed at pet owners. Children do not own animals, they can enjoy the company of the animal that their parents acquire as owners, the responsible for the pets are their parents.

Point 18: your new model of responsibility - that is a good term - but such a model has to start with education and cultural shifts? Have you read any work on responsive regulation and the role in any regulatory system of education and training? We cannot, can we go straight to regulation and enforcement when so many people are doing the right thing? Wouldn't it be better to spend money on identifying and educating those that do the wrong thing? Even work by elizabeth tyson on licensing laws and animal welfare might be a useful thing to consider here?

Response 18: This section has been deleted

Point 19: Your concept of s&h insurance. This is an interesting concept. But again, and I am sorry, I keep coming back to the economic realities of this. A wealthy businessman living in an apartment in new york, purchasing a $ 600o french bulldog might see the sense of such insurance and it might work here but what about a mouse owned by a small child in a village. A child that needs to learn about caring for animals as his family are farmers. Can we really see that mouse being insured? If so, can you put in some articles where we do now insurance has helped. Pet insurance does exist so let us start with looking at that. Put some detail about it in here and if there are articles that confirm it is a good thing and offers a viable business to anyone, put that detail in and expand

Response 19: Comment accepted. As far as we know, Compulsory Health and Survival Insurance (CHSI) scheme proposed in this article doesn’t exist in any country. We have changed lines  215-218. 

Point 20: Your comparison to the car industry insurance is a good one. The costs of pet care in a shelter can is not disputed. Would your argument, for this whole paper be better served by making a case, as you do here with dogs? At least some dogs do have enough value to warrant a consideration of making owners pay insurance, as i said about, many already do. Its too much to attempt to argue we can insure all animals. Perhaps propose it if for dogs in one or two countries and then make statements about the possibility of it moving into other animal breeding industries ?????

Response 20: This section has been deleted

Point 21: yes good points it might but not for all animals? Until all people value the lives of all animals equally, until we are educated about the value of animal life and until we stop being human centric what you have is a lovely philosophical position - not the framework for a viable insurance business or not the framework for a viable legal framework for the 195 countries in the world? If you have literature that disproves my view here, please pop it in.

Response 21: This section has been deleted

Point 22: I sound like the people your paper mentions indirectly (as always happens). I am not, I believe that the notion of insurance is a good one. It is the scope of your innovative idea - it needs to be smaller, its too big for 2020. Such a solution could be piloted with one animal and one country ??

Response 22: Actually, it will be the States insterested in implementing the project in their territory who will decide how many animals they want to join the project.

Point 23: I know in many countries and let us use Australia - many states, such as the state of Victoria have attempted to tackle overpoplation and abandonment for many years. I know you quote work from Australia so you will have researched it. Even in states that have attempted to tackle it, animals are still being abandoned. I think you need examples of what good has been obtained where legislation in the form that most closely aligns to what you are proposing has achieved? Can you find any?

Response 23: We have changed  lines  173-218 to clarify this section.

Point 24:yes, the creation of bodies that are there to protect animals and then to facilitate legislation at national level is a commendable thing - you need examples. Some countries have this, but can all countries really establish special protection and a push towards the emancipation of animals? Where is the political good will do this? Provide some examples where it does exist and go from there?

Response 24:  We have changed  lines  173-218 to clarify this section.

Point 25: Who will pay for the de sexing? There are studies that look at this are not there

Response 25: This section has been deleted

Point 26: You propose price setting but by whom? Previously in the paper you have advocated the use of market forces to help regulate animals. Now you propose some agency become involved. How will it work in reality?

Response 26: This section has been deleted

Point 27: You acknowledge no feasibility study has been carried out - but couldn't you at least propose a country where something as close to what you propose is occurring or a country that you believe might be able to have some of these measures considered?

Response 27: Comment accepted. We have added lines 220-235.

Reviewer 2 Report

None

Author Response

The authors thank the revisor for the time and effort employed in revising the article. We have improved English of the entirely manuscript, change the reference related to number of pet abandonments (lines 114-122) and clarifying the function and scope of the Compulsory Health and Survival Insurance (CHSI) scheme (line 195-218).

Reviewer 3 Report

The contribution addresses a very important topic, although in a way that sometimes can be confusing. Many concepts are introduced but not properly discussed. As presented, the proposal seems too much optimistic and abstract and it seems rather difficult to see how the system described can be enforced in the light of the current system. Additionally, when dealing with the issues of overpopulation and abandonment. the distinction between exotic pets and pets like dogs and cats would have to be discussed more in detail. Exotic pets trade, including illegal trade, is a distinct issue with specific consequences. In general, it would be appropriate to put the argument presentation more in order. The impression, on the part of the reader, could be that of an accumulation of many concepts and arguments, but not clearly distinguished and discussed. Language is sometimes too much generalist and not appropriate in english

Here are some general considerations:

  • Lines 30-34: too much generalist
  • Line 39: too much simplistic: two different issues that need to be addressed separately and in details
  • Lines 92-93: aims would have to be more precise, focusing on the specific contribution this study can give to the topic, not just saying that it will ‘end the suffering of abandoned pets’
  • Line 101: the numbering of this paragraph and of the remaining ones in the contribution would have to be changed. Are all the paragraphs in the contribution just part of the Introduction? And does the contribution contain just one more paragraph, that is the Conclusion, as paragraph 2?

Below some detailed suggestions:

  • Line 55: change the punctuation
  • Line 66: incidents, but in line 54 they are called ‘aspects’
  • Line 67-68: to be explained
  • Line 68: what does ‘operators’ mean in this context? Exotic pets and cats and dogs would have to be considered separately here and the specific issue of illegal trade of exotic pets would have to be added to the discussion as a separate one;
  • Line 82-83: too much simplistic
  • Lines 95-100 can be eliminated as redundant
  • Line 122: is B another type of ‘cost’ (see line 109) or a ‘problem’?
  • Line 122: it would have to be explained here what is meant with the affirmation that they cannot be rescued
  • Line 130: why the following sentence is at the head? there is already a bulleted list
  • Line 133: is faced
  • Line 134-135: too much simplistic. A distinct issue is here introduced which would need a separate discussion
  • Line 143: why the following sentence is at the head? there is already a bulleted list
  • Line 148-149: wild is not a synonymous of exotic
  • Line 153: this sentence does not seem correct: COVID-19 does not seem ’ to have been transmitted from pets to humans’
  • Line 175-176: reformulate. If overpopulation and abandonment have to be solved it is not ‘because..’ and it is not clear with ‘lines of research’ here means
  • Line 181-182: illegal trade is just one cause of the overpopulation issue
  • Line 186: archaic model: which one?
  • Line 187: philosophical doctrines: too much simplistic
  • Line 190-192: reformulate: it is not comprehensible to the reader
  • Line 197: in their favor them?
  • Line 197: this approach: which one? All the period 193-198 is rather obscure
  • Line 207: continue given?
  • Line 215: ‘ethical-moral’? why it is so called? And ‘ethical’ and ‘moral’ are synonymous, so why to use both?
  • Line 218-19: tangible material model?
  • Line 229: ‘could contribute to’ would be more realistic
  • Line 248: expected what?
  • Line 259: as note?
  • Line 273-74: ethical-moral? – same as before. Principles: which ones?
  • Line 275: same
  • Line 278-79: new model, new tools: too much generalist
  • Line 288: slaughter does not seems the right term to be used here
  • Line 290: what ‘general welfare’ is?
  • Line 299: which ones?
  • Line 319: which ones?
  • Line 331-32: redundant
  • Line 359: than the?
  • Line 360: all insurance?
  • Line 382: too much optimistic. What about pets which are not purchased, for example, or that are adopted? It has to be explained to the reader. It is not enough what is mentioned in lines 513ss
  • Line 395: depending on the type of pet, this proposal can have many detrimental effects on its welfare
  • Line 424ss: too much optimistic to be enforced
  • Line 489: there seem to be only two
  • Line 493: slaughtered: as above
  • Line 531-32: what this time unit?
  • Line 525: system

Author Response

The authors thank the revisor for the time and effort employed in revising the article. We have rewritten the entire manuscript to be more brief and precise. Below we answer each of your questions.

 Point 1: Lines 30-34: too much generalist

Response 1: Comment accepted. We have changed the abstract (lines 24 to 37).

Point 2: Line 39: too much simplistic: two different issues that need to be addressed separately and in details

Response 2: Comment accepted. We have changed the introduction section (lines 40 to 43).

Point 3 lines 92-93 aims would have to be more precise, focusing on the specific contribution this study can give to the topic, not just saying that it will 'end the suffering of abandoned pets'

Response 3: Comment accepted. We have rewritten the introduction section (lines 39-75).

Point 4: Line 101: the numbering of this paragraph and of the remaining ones in the contribution would have to be changed. Are all the paragraphs in the contribution just part of the Introduction? And does the contribution contain just one more paragraph, that is the Conclusion, as paragraph 2?

Response 4: Comment accepted. We have rewritten 2.1 section (lines 78-101).

 Point 5: line 55: change the punctuation

Response 5: This section has been deleted.

Point 6: line 66 incidents, but in line 70 they are called 'aspects'

Response 6: This section has been deleted.

Point 7 and 8:line 67-68 to be explained.What does 'operators' mean in this context? Exotic pets and cats and dogs would have to be considered separately here and the specific issue of illegal trade of exotic pets would have to be added to the discussion as a separate one;

Response 7 and 8

This section has been deleted.

Point 9: lines 82-83 Two much simplistic

Response 9: This section has been deleted.

Point 10: lines (95-100) can be eliminated as redundant

Response 10: This section has been deleted.

Point 11 and 12: Line 122, is B another type of 'cost' or a 'problem'? it would have to be explained here what is meant with the affirmation that they cannot be rescued

Response 11 and 12: Comment accepted. We have changed Lines 126 and 136. This means that unfortunately not all abandoned animals are rescued since it is not possible knowing its location.

Point 13: Line 130: why the following sentence is at the head? there is already a bulleted list

Response 13: Comment accepted. The section has been rewritten (lines 123-170).

Point 14: line 133 is faced, no face

Response 14: This section has been deleted

Point 15:lines 134-135 too much simplistic. A distinct issue is here introduced which would need a separate discussion

Response 15: Comment accepted. The section has been rewritten (lines 123-170).

Point 16:line 143 why the following sentence is at the head? there is already a bulleted list

Response 16: Comment accepted. The section has been rewritten  (lines 123-170).

Point 17: Lines 148-149 wild is not a synonymous of exotic

Response 17: Comment accepted, we have change wild for exotic (line 160).

 Point 18: line 153 this sentence does not seem correct: COVID-19 does not seem 'to have been transmitted from pets to humans'

Response 18:

Comment accepted. We have changed lines 164-166.

Point 19:lines 175-176 reformulate. If overpopulation and abandonment have to be solved it is not 'because ..' and it is not clear with 'lines of research' here means

Response 19: This section has been deleted.

Point 20:line 181-182, illegal trade is just one cause of the overpopulation issue.

Response 20:  Comment accepted. We have change lines 40-42.

Point 21: line 186 archaic model: which one?

Response 21: This section has been deleted

Point 22: line 187 philosophical doctrines: too much simplistic

Response 22: This section has been deleted

Point 23:lines 190-192 reformulate: it is not comprehensible to the reader

Response 23: This section has been deleted

Point 24: lines 197 In their favor them?

Response 24: This section has been deleted

Point 25:line 197 This approach: which one? All the period 193-198 is rather obscure

Response 25: This section has been deleted

Point 26: line 207 continue given?

Response 26: This section has been deleted

Point 27: line 215 ethical-moral '? why it is so called? And 'ethical' and 'moral' are synonymous, so why to use both?

Response 27: Comment accepted. We have changed the paragraph (line 99).

Point 28: lines 218-219, tangible material model?

Response 28: This section has been deleted

Point 29: lines 229 'could contribute to' would be more realistic

Response 29:  This section has been deleted

Point 30: Line 248 expected what?

Response 30: This section has been deleted

Point 31: line 259 as note?

Response 31: This section has been deleted

Point 32: lines 273-74 ethical-moral? - same as before. Principles: which ones?

Response 32: This section has been deleted

Point 33: line 275 same

Response 33: This section has been deleted

Point 34: lines 278-79 new model, new tools: too much generalist

Response 34: This section has been deleted

Point 35: line 288 slaughter does not seem the right term to be used here

Response 35: Comment accepted. We have employed sacrifice instead of slaughter in the entire manuscript

Point 36: lines 290 what 'general welfare' is?

Response 36: This section has been deleted.

Point 37: Line 299 which ones?

Response 37: This section has been deleted

Point 38: Line 319 which ones?

Response 38: Those described in lines 93-96.

Point 39: redundant lines 331-332

Response 39: This section has been deleted

Point 40: Line 359 than the?

Response 40: This section has been deleted

Point 41: Line 360 All insurance?

Response 41: Comment accepted. We hace rewritten this section (lines 195-218).

Point 42: line 382 too much optimistic. What about pets which are not purchased, for example, or that are adopted? It has to be explained to the reader. It is not enough what is mentioned in lines 513ss

Response 42: This section has been deleted

Point 43: line 395 Depending on the type of pet, this proposal can have many detrimental effects on its welfare

Response 43: Of course, this measure is thought to be implemented only in cases where characteristics of animals allow shared ownership.

Point 44: lines 424ss and ss too much optimistic to be enforced

Response 44: Comment accepted. We have rewritten 3.1 section (lines 173-194).

Point 45: line 489 there seem to be only two

Response 45: This section has been deleted

Point 46: line 493 slaughtered: as above

Response 46: Comment accepted. We have employed sacrifice instead of slaughter in the entire manuscript

Point 47: Line 531-32 what this time unit?

Response 47: This section has been deleted

Point 48: line 525 system

Response 48: This section has been deleted

Reviewer 4 Report

Dear Authors,

the paper addresses a very important topic such as that pet overpopulation and abandonment. The study is interesting and useful but really needs a lot more careful research for example about the influence of relevant regulatory frameworks in different countries.

In the text is not clear if the Authors address their attention at the Spanish law or that of other countries.

Significant revisions are required in the manuscript to warrant further consideration for publication of this manuscript in Animals

Please, find below some specific points that the authors may change:

Line 12: I believe that the sanctions are sufficient instruments. Where they are insufficient? It is not specified

Lines 23 and 24: You wrote: “Overpopulation and abandonment of pets is a historical and endemic problem in many if 
not most countries of the world that remains unsolved”. In which countries of the world? It is not specified

Line 44: Which law? It is not specified

Lines 55-56: Point A – In which country? Not all countries of EU euthanize abandonment pets

Line 62; Point C – Please add reference

Lines 199-202, 205: This is not always the case. It depends on the country. More specifically, one has to clarify what country you are talking about. The text is confusing. In some countries the microchip is mandatory, in this manner the abadonment is not anonymous.

Lines 244-245: You cannot generalize, because there are countries where there are controls or verifications before the purchase of a pet.

Lines 285-286: Are you sure? I think that there are countries that are resolved these problems (i.e. Switzerland).

Line 337: Which law? In which country?

Line 469: There are countries that regulate an effective model of responsible pet ownership.

Author Response

The authors thank the revisor for the time and effort employed in revising the article. The objective of the article is not to make a detailed analysis of the legislation of each country in the world but of the common aspects they share in relation to the management of overpopulation and abandonment. We have also rewritten the entire manuscript to be more brief and precise. Below we answer each of your questions.

Point 1: line 12 I believe that the sanctions are sufficient instruments. Where they are insufficient? It is not specified

Response 1: Comment accepted. We have added lines 81-84.

Point 2: lines 23-24 You wrote: "Overpopulation and abandonment of pets is a historical and

endemic problem in many if not most countries of the world that remains unsolved”; In which countries of the world? It is not specified  

Response 2: This section has been rewritten (lines 40-42).

 Point 3: line 44 Which law? It is not specified

Response 3: Comment accepted. We have change it for lines 103-108 to clarify it (is a basic economic principle).

Point 4: line 55-56 In which country? Not all countries of EU euthanize abandonment pets

Response 4: Euthanasia is a common practice when abandoned pets accumulate unmanageable (references 5-9 and 10,11).

Point 5: line 62 point C Please add reference

Response 5: Comment accepted. Introduction section has been rewritten (lines 39-75).

Point 6: lines 199-202, 205 This is not always the case. It depends on the country. More specifically, one has to clarify what country you are talking about. The text is confusing. In some countries the microchip is mandatory, in this manner the abadonment is not anonymous.

Response 6: Comment accepted. We have added lines 85-90.

Point 7: lines 263-267 You cannot generalize, because there are countries where there are controls or verifications before the purchase of a pet.

Response 7: Coment accepted. As far as we know, no country has a comprehensive managament system for pet overopulation and abandonment like the one proposed in the article. Section 3.1 has been rewritten (lines 173-218).

Point 8: lines 307-308 Are you sure? I think that there are countries that are resolved these problems (ie Switzerland)

Response 8: This section has been deleted.

Point 9:Line 362: Which law? In which country?

Response 9: Thank you for your comment. It refers to '' Law '' in the institutional sense, not to a particular law.

Point 10: Line 509: There are countries that regulate an effective model of responsible pet ownership.

Response 10:  Coment accepted. As far as we know, no country has a comprehensive managament system for pet overopulation and abandonment like the one proposed in the article. We have rewritten lines 173-218.

Round 2

Reviewer 1 Report

Ok - this paper is now written in a better way, it sets out what it says it will, a novel approach to helping with abandoned pets.

I am content now, if the editors are with the publication of this paper. 

Author Response

The authors thank the reviewer for the time spent to review the article and the comments made to improve it.

Kind regards,

Eva B.

Reviewer 2 Report

I appreciate the opportunity to review the changes made to this manuscript and the time the authors have taken to address some of my concerns. The article is now much more readable in terms of English and therefore the points and arguments raised in the paper will be more accessible to a wider audience.

As written, I believe this paper is an opinion piece but there is significantly more that can be done to strengthen this opinion. The authors have presented an argument in favour of compulsory insurance to help reduce the impact of unwanted pets on society and the planet. The principle itself is sound but the realities of implementing such a scheme are highly problematic and these have not been sufficiently discussed. The paper would be significantly more impactful if the authors were to perform an in depth economic and viability assessment for a specific country where the variables are understood, the practicalities can be specified, and the impact on the pet population more accurately modelled. This in itself would represent a substantial piece of research and if published would make a significant contribution to the debate. It would also allow others to consider the principle, assumptions and impact for other countries where the circumstances may be similar or substantially different. For example, is this a viable option in some European countries with existing animal ownership regulations, versus the USA where there are local state cultural and regulatory differences. Furthermore, how might such a scheme operate in countries such as India where the concept of shelters is not part of the culture.

I would encourage the authors to consider this economic modelling research which, if done well, could positively impact animal wlefare.

Author Response

The authors thank the reviewer for the time spent in revising the manuscript and the comments made.

In relation to your question, we comment the following:

Taking into account that no country in the world has a comprehensive management system to address overpopulation and abandonment with the characteristics described in this manuscript, it is not possible to carry out an economic and feasibility evaluation of its implementation a priori, because it is necessary to agree with each country directly the guidelines and contents of what each government wants to obtain and the way they want to achieve it (creation of ad hoc regulations, support from the different powers of the state, management protocols, technical and financial support available, the infrastructures to be used, the necessary human resources, etc.).

Each country has specific political, economic, religious and cultural circumstances. They need a differentiated treatment. The result of what is agreed, taking into account all these variables, can be very different and therefore the economic and feasibility evaluation will be very different in each case. What is proposed in this article is a comprehensive management system for overpopulation and abandonment of companion animals, which can be technically applied in any country in the world as long as governments have a real interest in solving the problem of overpopulation and abandonment of pets. Obviously, there are countries where this management system is more feasible to establish and implement than others. For example, in India, given its characteristics, it would be much more complex to implement the project than in countries with extensive legislation and animal-friendly culture (from a Western perspective), such as the United Kingdom, but technically the system can be applied to any country in the world as long as governments understand that the country really needs to solve the problems of overpopulation and abandonment of pets, to end the precariousness of these animals and also the negative externalities that this situation entails.

A possible solution would be to put the project into operation in a place where it is easier to implement it (such as the United Kingdom), to serve as an example to demonstrate its feasibility and the advantages of its application. This will break the barrier of uncertainty generated by a novel project of these characteristics, and will make it easier for other countries to encourage to implement it.

Reviewer 3 Report

In general, the content of the manuscript has been ameliorated and can be promising, but is still to be improved. Lines 72-75 are promising too much to the reader: the rest of the contribution is not offering the reader any comprehensive discussion, but just some hints, of what promised. It is suggested also to check for english and punctuation

Below some detailed suggestions:

Line 2: instead of ‘Amelioration’ it is suggested mitigation or something similar

Line 20: same

Line 21: as well as to   same 36

Line 22: flexible: the reader is not provided with an explanation of this flexibility throughout all the manuscript

Line 22: a scheme, an instrument or a protocol? Obscure. See section 3.2

Line 28ss: problems concerning ecological what?; the same for economic

Line 33: same as 2 and 20

Line 36: these approaches or combination of approaches

Line 63: )

Line 65: the whole paragraph forces the reader to try to distinguish between consequences, factors and Issues. It is suggested to reformulate it more clearly to help the reader appreciate the content

Line 67: the concept of intrinsic value would deserve an explanation

Line 75: to mitigate these problems?

Line 77: Discussion of what? It is suggested to reformulate. The whole section is not a discussion

Line 81: ‘measure’ or mechanism?

Line 85: provides or prescribe?

Line 89: there was some misunderstanding?

Line 93ss: the reader is confused, since the connection with the previous sentence is not clear

Line 97: which one? The whole point 3 in its present form is obscure

Line 102: why 2.4 after 2.1.?

Line 109 ss: the sentence is obscure and the concept introduced would deserve an explanation

Line 172: Before to proceed, the reader would need to be introduced to the various paragraphs of this section, after been presented with just a list of issues, consequences, factors etc. in the previous section

Line 174: too much simplistic: this can be one factor among others

176-179: obscure

Line 180-187: too much content in one paragraph, without properly unpacking the different issues

Line 188-194: same

Line 195ss: the reader deserves a more detailed explanation of the content of the whole section

Lines 220-222: too much simplistic

Line 236ss: the conclusion does not help the reader to conclude. Instead generates more confusion

Line 241: which events?

Line 242: training courses: which ones?

Author Response

The authors thank the reviewer for the time spent in reviewing the article and the comments made to improve it. Regarding the general comment on the article, we have shortened the length of the manuscript according to the journal's requirements and we have changed line 73 according to your comment. Below we answer each of your questions.

Line 2: instead of ‘Amelioration’ it is suggested mitigation or something similar
The word ‘amelioration’ means to improve a situation, thus it does seem appropriate.

Line 20: same
The word ‘amelioration’ means to improve a situation, thus it does seem appropriate.

Line 21: as well as to   same 36
Comment accepted. We have changed lines 21 and 36.

Line 22: flexible: the reader is not provided with an explanation of this flexibility throughout all the manuscript
Comment accepted. We have changed line 36.

Line 22: a scheme, an instrument or a protocol? Obscure. See section 3.2
Comment accepted. According to line 21, we have changed instrument for measure (line 202), and the second term ‘scheme’ has been deleted (line 204).

Line 28ss: problems concerning ecological what?; the same for economic
The section has been written according to the requirements of the journal.
Line 33: same as 2 and 20
The word ‘amelioration’ means to improve a situation, thus it does seem appropriate.

Line 36: these approaches or combination of approaches
The section has been written according to the requirements of the journal.

Line 63: )
Comment accepted. We have added it.

Line 65: the whole paragraph forces the reader to try to distinguish between consequences, factors and Issues. It is suggested to reformulate it more clearly to help the reader appreciate the content
The section has been written according to the requirements of the journal.

Line 67: the concept of intrinsic value would deserve an explanation
Comment accepted. The correct term is value

Line 75: to mitigate these problems?
The section has been written according to the requirements of the journal.

Line 77: Discussion of what? It is suggested to reformulate. The whole section is not a discussion
The section has been written according to the requirements of the journal.

Line 81: ‘measure’ or mechanism?
Comment accepted. We have changed line 80

Line 85: provides or prescribe?
Comment accepted, prescribe is probably a better term

Line 89: there was some misunderstanding?
Comment accepted, we have changed line 90

Line 93ss: the reader is confused, since the connection with the previous sentence is not clear
Comment accepted, we have changed line 93-103

Line 97: which one? The whole point 3 in its present form is obscure
Comment accepted, we have changed line 93-103

Line 102: why 2.4 after 2.1.?
Comment accepted, we have changed it, is section 2.2

Line 109 ss: the sentence is obscure and the concept introduced would deserve an explanation
Comment accepted. We have changed line 105, 110 and 111.

Line 172: Before to proceed, the reader would need to be introduced to the various paragraphs of this section, after been presented with just a list of issues, consequences, factors etc. in the previous section
The section has been written according to the requirements of the journal

Line 174: too much simplistic: this can be one factor among others
The section has been written according to the requirements of the journal. Anyway we have added basically in line 177 according to your comment.

Lines 176-179: obscure
The section has been written according to the requirements of the journal.

Line 180-187: too much content in one paragraph, without properly unpacking the different issues
Comment accepted. We have split paragraph (lines 177-199).

Line 188-194: same
Comment accepted. We have split paragraph (lines 202-224).

Line 195ss: the reader deserves a more detailed explanation of the content of the whole section
The section has been written according to the requirements of the journal.

Lines 220-222: too much simplistic
The section has been written according to the requirements of the journal. We have changed lines 230 and 236-239.

Line 236ss: the conclusion does not help the reader to conclude. Instead generates more confusion
Comment accepted. We have changed lines 241-253.

Line 241: which events?
Comment accepted. We have changed line 244-245

Line 242: training courses: which ones?
Comment accepted. We have changed line 246-247

Reviewer 4 Report

Dear Authors,

made revisions have improved your manuscript considerably.

I recommend the publication of the manuscript in Animals.

Author Response

(The authors gave the same response as above.)
